# Gut mycobiome alterations and network interactions with the bacteriome in patients with atherosclerotic cardiovascular disease

Guangming Su,[1] Ping Huang,[2] Dan Liu,[1] Guorui Xing,[3] Ruochun Guo,[3] Shenghui Li,[3] Shao Fan,[4] Lin Cheng,[4] Qiulong Yan,[4,5] Wei Yang[1]

**ABSTRACT**    The connection between the gut mycobiome and atherosclerotic cardiovascular disease (ACVD) is largely uncharted. In our study, we compared the gut fungal communities of 214 ACVD patients with those of 171 healthy controls using shotgun metagenomic sequencing and examined their interactions with gut bacterial communities and network key taxa. The gut mycobiome composition in ACVD patients is significantly different, showing a rise in opportunistic pathogens like *Candida albicans*, *Exophiala spinifera*, and *Malassezia restricta*, with *Exophiala* and *Malassezia* showing the most significant changes (Wilcoxon rank-sum test, $P < 0.001$, fold change >10). Network analysis revealed a less interconnected and more uneven gut microbial network in ACVD patients. Network key taxa identified in the ACVD gut microbiome network include *Malassezia globosa* c182, *Nakaseomyces glabratus* c88, *Malassezia arunalokei* c192, and *Penicillium sumatraense* c22. Predictive models that integrated both bacterial and fungal taxa enhanced prediction accuracy, underscoring the critical role of gut fungi in ACVD. Our findings offer a thorough understanding of the link between the gut mycobiome and ACVD progression, which is vital for directing future therapeutic research.

**IMPORTANCE**    ACVD is a leading cause of death and morbidity worldwide. While the role of the gut microbiome in ACVD development is recognized, the contribution of the gut mycobiome remains largely unexplored. Our study reveals significant alterations in the gut mycobiome of ACVD patients and identifies key fungal taxa associated with the disease. These findings underscore the importance of the gut mycobiome in the pathogenesis of ACVD and offer new avenues for developing preventive and therapeutic strategies targeting the gut fungal community. Our results provide valuable insights into the complex interplay between gut fungi and bacteria in ACVD, paving the way for novel therapeutic approaches.

**KEYWORDS**    atherosclerotic cardiovascular disease, gut mycobiome, gut bacteriome, shotgun metagenome sequencing, network interaction

Atherosclerotic cardiovascular disease (ACVD) is a major global health challenge, accounting for significant morbidity and mortality worldwide, affecting millions of individuals each year (1). ACVD includes conditions such as coronary artery disease, stroke, and peripheral artery disease (2), primarily caused by the accumulation of atherosclerotic plaques in the arteries. These plaques, formed by lipid deposits, inflammatory cells, and fibrous tissue, narrow the arteries and restrict blood flow, leading to serious cardiovascular events (3). The pathogenesis of ACVD is complex, involving factors like lipid metabolism, systemic inflammation, and immune responses (4, 5). Inflammation, in particular, plays a crucial role in the progression of ACVD (6).

**Peer Reviewer** Mushtak T. S. Al-Ouqaili, University of Anbar, College of Medicine, Ramadi, Al-Anbar, Iraq

Address correspondence to Qiulong Yan, Qiulongy1988@163.com, or Wei Yang, yangwei6295@163.com.

Guangming Su, Ping Huang, Dan Liu, and Guorui Xing contributed equally to this article. The author order was determined by the corresponding author after negotiation.

The authors declare no conflict of interest.

Recent research has underscored the significant impact of the gut microbiome on cardiovascular health (7). The gut microbiome, a diverse community of microorganisms including bacteria, fungi, viruses, and archaea, plays a crucial role in maintaining human health by influencing metabolic functions, immune responses, and even behavior (8, 9). Dysbiosis of the gut microbiota contributes to the pathogenesis of ACVD by promoting systemic inflammation, metabolic dysfunction, and endothelial impairment, such as Enterobacteriaceae, *Streptococcus* spp., and Lachnospiraceae (10). Conversely, beneficial bacteria such as *Bifidobacterium* and certain species of *Lactobacillus* are often depleted in these patients, which may exacerbate the disease process by weakening the gut barrier and increasing inflammatory responses (11).

While some studies have explored the relationship between gut microbiota and ACVD, most research has primarily focused on the bacterial components of the gut microbiome. Emerging evidence indicates that the gut microbiome, including its fungal inhabitants, plays a significant role in the onset and progression of various human diseases (12). Although the gut mycobiome is less abundant than bacteria, it can have a notable impact on host physiology. A recent preliminary study suggested that gut fungi may be linked to carotid atherosclerosis, identifying certain species from the Mucoraceae family and the *Mucor* genus as potential contributors to cardiovascular diseases. Additionally, fungi such as *Candida* and *Malassezia* are known to influence immune responses and inflammatory pathways (13). Changes in gut fungal populations are associated with systemic inflammation, a key factor in the development of ACVD. Therefore, understanding the specific roles and interactions of gut fungi in ACVD could provide new insights into disease mechanisms and identify potential therapeutic targets. This highlights the need for systematic research into the composition and interactions of the gut mycobiome in relation to this disease.

In this study, we performed a metagenomic-based characterization of the gut fungal community in patients with ACVD. The fecal metagenome data set was downloaded from a previous study on a total of 214 ACVD patients and 171 healthy controls (HCs). The gut mycobiome was profiled from fecal metagenomes and compared between patients and healthy controls, revealing numerous fungal compositional and functional signatures associated with ACVD. Moreover, the ability of fungal signatures to classify ACVD patients and healthy controls was also explored.

## MATERIALS AND METHODS

### Subjects and data set

The fecal metagenomic data set of 385 samples from 214 ACVD patients and 171 healthy volunteers was downloaded from the European Bioinformatics Institute database under the accession code ERP023788 . All patients were ethnic Han Chinese with no known consanguinity, aged 40–80 years old. The exclusion criteria included ongoing infectious diseases, cancer, renal, or hepatic failure, peripheral neuropathy, stroke, as well as use of antibiotics within 1 month of sample collection. All the healthy control individuals enrolled were free from clinically evident ACVD symptoms at the time of the medical examination. Demographic data and cardiovascular risk factors were collected by a questionnaire. Individuals with peripheral artery disease, known coronary artery disease or myocardial infarction, cardiomyopathy, renal failure, peripheral neuropathy, systemic disease, and stroke were excluded. Additionally, none of the ACVD patients had received steroids or antibiotics within the preceding 3 months. Patients with ACVD exhibited a significantly lower proportion of females (25.2%) compared to the healthy control group (59.4%, Table S1). However, no significant differences were observed in terms of age (61 $\pm$ 10 years for patients vs 60 $\pm$ 10 years for healthy controls; Student's $t$-test, $P = 0.548$) or body mass index (BMI) (24.6 $\pm$ 3.5 vs 24.5 $\pm$ 6.8; Student's $t$-test, $P = 0.842$) between the two groups.

## DNA extraction from fecal samples and DNA library construction

Fecal samples were thawed on ice, and DNA extraction was performed using the Qiagen QIAamp DNA Stool Mini Kit (Qiagen) according to manufacturer's instructions. Extracts were treated with DNase-free RNase to eliminate RNA contamination. DNA quantity was determined using NanoDrop spectrophotometer, Qubit Fluorometer (with the Quant-iTTMdsDNA BR Assay Kit), and gel electrophoresis. DNA library construction was performed following the manufacturer's instruction (Illumina). We used the same workflow as described previously to perform cluster generation, template hybridization, isothermal amplification, linearization, blocking and denaturation, and hybridization of the sequencing primers. We constructed a paired-end (PE) library with an insert size of 350 bp for each sample, followed by high-throughput sequencing with PE reads of length $2 \times 100$ bp.

## Construction of gut fungi genome catalog

The available National Center for Biotechnology Information fungal genomes were downloaded in June 2024. The raw fungal genomes included 16,634 genomes, 1,384 of which were removed because of (i) extremely low assembly quality (N50 length <2,000 bp or number of scaffolds >10,000) or (ii) a mixture of multiple genomes, with the remaining 15,250 genomes retained as a reference for further analyses. Meanwhile, to ensure accurate fungal identification, we aligned the fungal genomes with the bacterial sequences in the NT database and removed any sequences matching the bacteria. Incorporating methods from published articles (14), we updated the original database and identified a total of 1,490 human-associated fungal species. These strains were then clustered into 317 non-redundant human-associated fungal species using an average nucleotide identity (ANI) threshold of 95%.

## Processing of metagenomic sequencing data

To ensure data quality, we utilized fastp v.0.20 to process each metagenomic sample. The raw reads underwent several quality control steps, including the trimming of polyG tails and the removal of low-quality reads based on the following criteria: (1) reads shorter than 90 bp; (ii) reads with a mean Phred quality score below 20; (3) reads with more than 30% of bases having a Phred quality score below 20; (4) reads with a mean complexity under 30%, such as repetitive sequences like "ATATATATAT" or homopolymeric stretches like "AAAAAAA," which have low complexity scores; and (5) unpaired-end reads.

The gut bacteriome composition of fecal samples was profiled based on the extensive Unified Human Gastrointestinal Genome (UHGG) database (15). The metagenomic reads for samples of ACVD patients and HC groups were aligned against the UHGG database to generate the gut bacteriome profiles. Reads that mapped to the bacterial rRNA/tRNA gene sequences were dismissed. Relative abundances of 899 prokaryotic species were calculated by normalizing for each sample, and the relative abundances at the phylum and genus levels were obtained by summing the abundances of species from the same taxa.

To minimize the impact of non-specific mapping of reads to fungal genomes in subsequent analyses, we mapped the filtered reads against three databases: the GRCh38 genome, the UHGG collection, and the SILVA rRNA database (16). This step enabled us to exclude reads originating from human or prokaryotic sources. For each sample, the remaining reads were aligned against our customized catalog of gut fungal genomes using bowtie2, and the read counts for each genome were calculated. To generate mycobiome composition profiles, we employed a multi-step normalization process to ensure accurate determination of relative abundances. Initially, the read count for each genome was normalized by dividing it by its genomic size. This step was crucial to account for variations in genome sizes across different species and to prevent bias in abundance estimations. Following this, the normalized read count for each genome was further processed using the transcript per million (TPM) approach. In the TPM method,

the normalized read count of each genome was divided by the sum of all normalized read counts in a sample then multiplied by one million. This approach ensures that the sum of all relative abundances in a sample equals one million, facilitating comparison across samples. For different fungal taxa, the relative abundance of a taxon was calculated as the sum of the relative abundances of all populations assigned to that taxon. This process preserved the relative abundance of each population within the sample, allowing for accurate profiling of the mycobiome composition.

## Statistical analyses and visualization

We determined the number of observed species by counting those with a relative abundance greater than zero in each sample. Shannon's index and Richness's index were calculated using the "diversity" function in the *vegan* package. A Bray–Curtis distance matrix was created using the square-root transformed species-level profiles, employing the *vegdist* function from the vegan package (17). Principal coordinate analysis (PCoA) was subsequently conducted on the distance matrix using the *pcoa* function in the ape package. To assess the variance, permutational multivariate analysis of variance (PERMANOVA) was performed on the distance matrix with the *adonis* function in the *vegan* package. The Wilcoxon rank-sum test was implemented using the function *wilcox.test*. Student's *t*-test was implemented using the function *t.test*. Linear discriminant analysis (LDA) scores measure group separation by maximizing the distance between group means and minimizing within-group variation. Fold change (FC) is calculated as the ratio of species abundance between conditions. We used LDA (>2) and FC (>1.2) to detect and filter species, ensuring accurate identification of significant microbial differences and enhancing result reliability. The sunburst diagram of taxonomic hierarchy was generated using the function *plot_ly* in the package *plotly*. The phylogenetic tree was built using PhyloPhlAn (18) and visualized in iTOL (19). All other data were visualized using the function *ggplot* in the package *ggplot2*.

To investigate the differences in interaction patterns between fungi and bacteria in the gut of healthy individuals and patients with ACVD, we employed network methods based on random matrix theory (RMT) to construct two co-occurrence networks using the platform available at http://ieg2.ou.edu/MENA. The RMT method ensures that the association strength adheres to a Poisson distribution under natural conditions. Additionally, we used Inference of Direct and Indirect Relationships with Effective Copula-based Transitivity to eliminate potential spurious indirect connections in the original network, including pathological connections, self-loops, and overly strong interactions within the ecological network (20). To minimize false positivity, only species detected in more than 10% of biological replicates for each group were included. We focused on the interactions among taxa involved in bacteria and fungi. Cytoscape v.3.8.2 software was used to visualize the networks (21).

The random forest (RF) classifier based on the gut mycobiome was constructed using the *randomForest* function, followed by five repetitions of tenfold cross-validation, with patient samples in each fold representing one-tenth of their respective total sample sizes. This ensured that the class distribution remained consistent within each fold. The performance of the classifier was evaluated based on the area under the receiver operating characteristic curve, which was calculated using the *roc* function. The importance ranking of the markers was obtained using the *importance* function. Additionally, the least absolute shrinkage and selection operator (LASSO) models were developed using the LASSO function followed by five times of fivefold cross‐validations, and their performances were assessed based on area under the curve (AUC) that was calculated via the roc function.

## RESULTS

### Sample information and fungal database

To characterize the gut mycobiome community in patients with ACVD, we analyzed the metagenomic sequencing data set from fecal samples of 214 patients and 171 healthy controls. To accurately define the mycobiome's composition, we used a tailored fungal database utilizing strict filtering criteria (see Materials and Methods for details). This refined database includes 317 distinct reference species, categorized based on a 95% ANI criterion, derived from an assortment of 1,490 genomes linked to the human microbiome (Table S2). Subsequently, high-quality reads from each sample were aligned to the genomes of these 317 distinct species present in our database, facilitating the creation of in-depth mycobiome profiles. Furthermore, our examination revealed the presence of 178 high-level taxa across the samples, which comprised 84 genera, 49 families, 27 orders, 14 classes, and 5 subphyla (Fig. 1). To determine the effects of gender, age, and BMI on the gut mycobiome, we utilized PERMANOVA to compare the impacts of these three indicators. The findings revealed that none of the factors—gender (Adonis = 0.0043, $P$ = 0.169), age (Adonis = 0.0035, $P$ = 0.171), or BMI (Adonis = 0.0037, $P$ = 0.129)—significantly affected the structure of the gut fungal community.

### Altered gut mycobiome structure in ACVD patients

We compared the gut mycobiome composition between healthy controls and ACVD patients using PCoA and PERMANOVA. Bray–Curtis distance-based PCoA of species-level composition revealed that PCoA1 and PCoA2 accounted for 18.4% and 10.4% of the total variation, respectively (Fig. 2A). ACVD patients showed a mild but statistically significant separation from healthy controls along PCoA1 (Wilcoxon rank-sum test, $P$ < 0.05). PERMANOVA confirmed significant differences in the gut mycobiome between the groups (Adonis = 0.014, $P$ < 0.001). Alpha diversity, assessed using richness and Shannon indices, indicated higher mycobiome richness in ACVD patients compared to healthy controls (Wilcoxon rank-sum test, $P$ < 0.001, Fig. 2B), while Shannon's index showed no significant difference between the groups.

In terms of the fungal taxa, the gut mycobiome of all subjects was usually dominated by Saccharomycotina, followed by Pezizomycotina, Basidiomycota, and Mucoromycota (Fig. 2C). At the genus level, *Penicilium* was the first most abundant genus, while other common genera, such as *Pichia*, *Barnettozyma*, and *Barnettozyma*, had relatively high abundances in both groups (Fig. 2D). According to the comparison analysis, a total of 14 genera exhibited significant differences (FC >1.2, relative abundance >0.01; Table S3) between ACVD patients and healthy controls (Fig. 2E). Among these genera, two stood out with the most prominent differences (FC >10), namely, *Exophiala* (Wilcoxon rank-sum test, adjusted $P$ < 0.001) and *Malassezia* (adjusted $P$ < 0.001).

### Gut fungal signatures associated with ACVD

Based on the significant differences observed at the genus level, we further investigated the differential taxa at the species level, aiming to identify potential biomarkers associated with ACVD. To achieve this, we applied stringent filters, considering an FC of >1.2, an LDA of >2, and an average relative abundance greater than 0.01. The analysis revealed a total of 25 significantly different fungal taxa, spanning across 7 classes and 13 genera (Fig. 3; Table S4). At the class levels, five fungal taxa were significantly enriched in ACVD patients, including the classes Saccharomycetes, Sordariomycetes, Malasseziomycetes, Tremellomycetes, and Agaricomycetes. At the genus level, eight fungal populations, including *Candida*, *Malassezia*, *Exophiala*, *Nakaseomyces*, *Bamettozyma*, *Microascales*, *Cryptociccus*, and *Irpex*, were significantly enriched in ACVD patients, whereas *Rhizopus* and *Trichophyton* were signifcantly enriched in healthy controls. At the species level, 20 fungal species were enriched in ACVD patients, while five species were enriched in healthy controls. Particularly noteworthy was the significant increase in the gut fungus *Candida albicans* among ACVD patients compared to healthy controls. In addition,

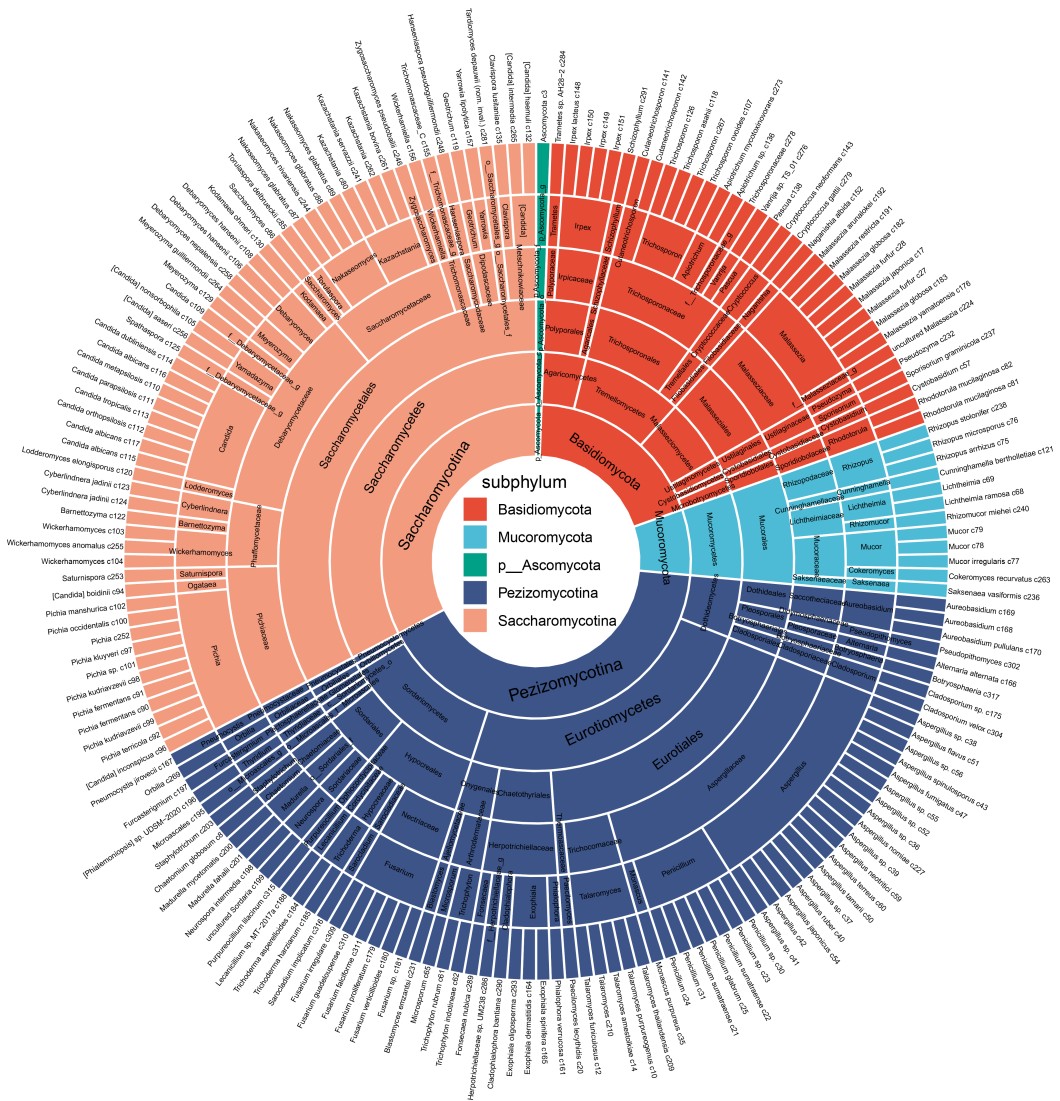

**FIG 1** Sunburst diagram of taxonomic hierarchy for 317 gut fungal species and 178 high-level taxa.

various opportunistic pathogens, including *Aspergillus* sp., *Exophiala spinifera*, *Malassezia restricta*, *Candida albicans*, *Mucor* sp., and *Malassezia furfur*, were found to be enriched in ACVD patients.

## Comparative analysis of gut microbiome co-occurrence networks

Due to the co-existence of multiple microorganisms in the gut, including bacteria and fungi, which actively collaborate and collectively form the gut microbiota network, it is crucial to further investigate the interplay among the gut microbiome and mycobiome. We applied uniform criteria to identify bacterial species that exhibited significant differences between individuals with ACVD and healthy individuals, utilizing these differentially abundant taxa to construct a gut microbiota network (Table S5). By excluding non-differential species, we aimed to streamline the network structure, emphasizing the influence of the disease on the intricate interplay among gut microbiota and facilitating the identification of potential key microbial players. In comparing the network topology characteristics between the ACVD group and the healthy control group, significant differences emerged. Despite having fewer nodes, the healthy control group exhibited a slightly higher number of edges, indicating a more interconnected network (Fig. 4). This resulted in a denser and more efficient network structure,

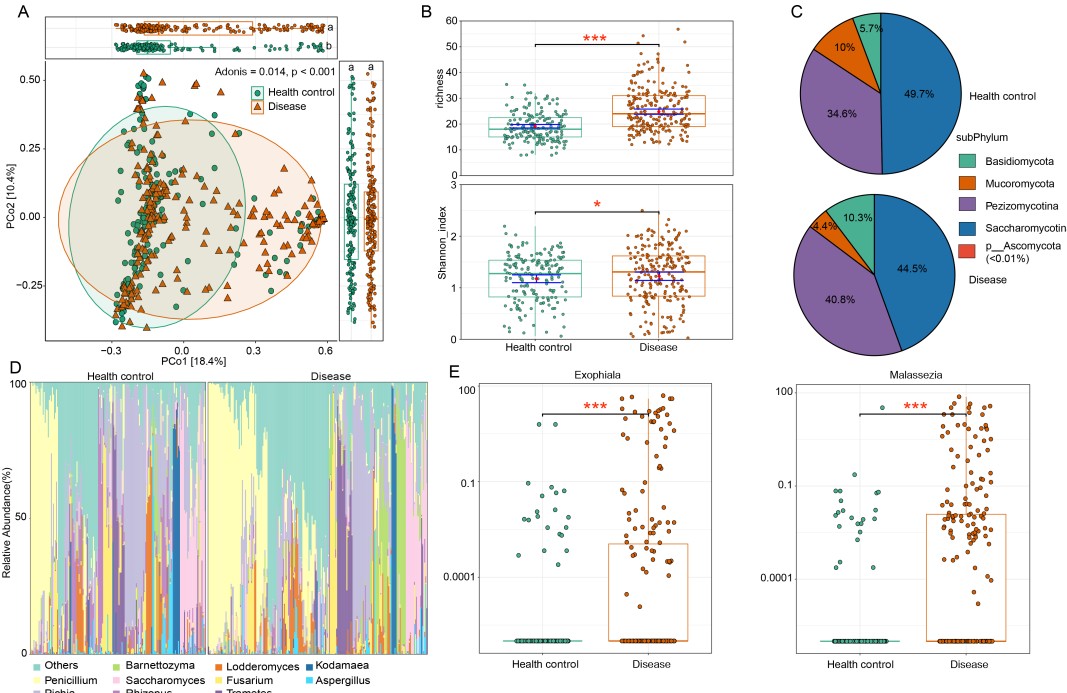

**FIG 2** Comparison of gut mycobiome diversity and structure between ACVD patients and healthy controls. (A) PCoA based on Bray–Curtis distance of the fungal profiles at the species level. The plot displays the distribution of samples along PCoA1 and PCoA2, with ellipsoids indicating the 95% confidence interval for each group. The upper and right boxplots display the sample scores in PCoA1 and PCoA2. (B) Comparison of alpha diversity indexes between ACVD patients and healthy controls. The *P* value was determined by the Wilcoxon rank-sum test. (C) Pie chart showing the composition of fungal subphyla in each group. The percentages represented the average relative abundance of each subphylum. (D) Distribution of the top 10 abundant genera across all samples. (E) Boxplots showing the relative abundances of the genera *Exophiala* and *Malassezia* in each group (FC >10, relative abundance >0.01). Statistical significance was determined using the Wilcoxon rank-sum test with Benjamini and Hochberg adjustment. *$P < 0.05$, ***$P < 0.001$.

characterized by a greater average number of neighbors and a shorter characteristic path length. Additionally, the healthy control group showed a lower network heterogeneity and centralization, suggesting a more uniform distribution of connections without prominent central nodes (Table S6). To delve deeper into the specific taxa involved, we examined the top 10 high-degree nodes in each group. In the healthy control group, these central nodes were predominantly composed of bacterial taxa. In contrast, the ACVD group demonstrated a significant increase in fungal taxa among the central nodes, including *Malassezia globosa* c182, *Nakaseomyces glabratus* c88, *Malassezia arunalokei* c192, and *Penicillium sumatraense* c22.

In summary, the healthy control group displayed a denser and more efficiently connected network with a uniform distribution of connections, indicating enhanced information transfer efficiency. Conversely, the ACVD group presented an altered network structure with an increased presence of fungal taxa among the central nodes. These findings provide valuable insights into the specific microbial structure within the gut microenvironment related to ACVD.

## Classification of ACVD state based on the gut microbiome

Finally, classification models were built using two machine learning algorithms (i.e., RF and LASSO) followed by five times of tenfold cross‑validations, and their performances were assessed by calculating the AUC. We employed a random forest classification model with fivefold cross-validation using the relative abundances of gut bacterial and fungal profiles. The results indicated that the bacterial and fungal models achieved cross-validated AUCs of 0.861 (95% confidence interval [CI]: 0.836–0.886) and 0.810 (95% CI: 0.791–0.829), respectively (Fig. 5A). The bacterial model was more effective

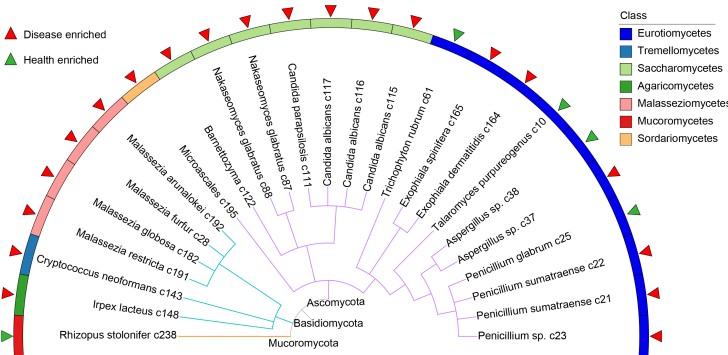

**FIG 3** The taxonomic tree of all differentially enriched species. Each tree-like branch with a different color represents a single phylum. The bar chart outside the fan chart describes the class level classification of a particular species. Species that were enriched in ACVD patients are shown as red triangles, and depleted species are shown as green triangles.

in distinguishing ACVD patients from healthy controls compared to the fungal model. Notably, combining bacterial and fungal data for joint prediction using 80 taxa yielded a higher cross-validated AUC of 0.868 (95% CI: 0.853–0.883) and significantly increased sensitivity compared to single-taxa predictions (Fig. 5C). Compared to the RF model, the LASSO model based on fungal samples showed a slightly lower predictive performance for distinguishing between healthy and diseased groups, with an AUC value of 0.803. However, the performance of the LASSO model improved when using bacterial samples (0.909) and when combining both fungal and bacterial samples (0.900). While bacterial species predominated in the prediction models of both methods, several fungal species—such as *Rhizopus stolonifer* c238, *Aspergillus* sp. c38, *Malassezia restricta* c191, *Penicillium* sp. c23, and *Barnettozyma* c122—exhibited high discrimination importance in the random forest model (Fig. 5B).

## DISCUSSION

In this study, we developed a comprehensive gut fungal genome database closely associated with the human mycobiome. Using this database, we conducted an in-depth metagenomic analysis of the mycobiome in fecal samples from 214 ACVD patients and 171 healthy controls. To our knowledge, this is the first investigation of the gut

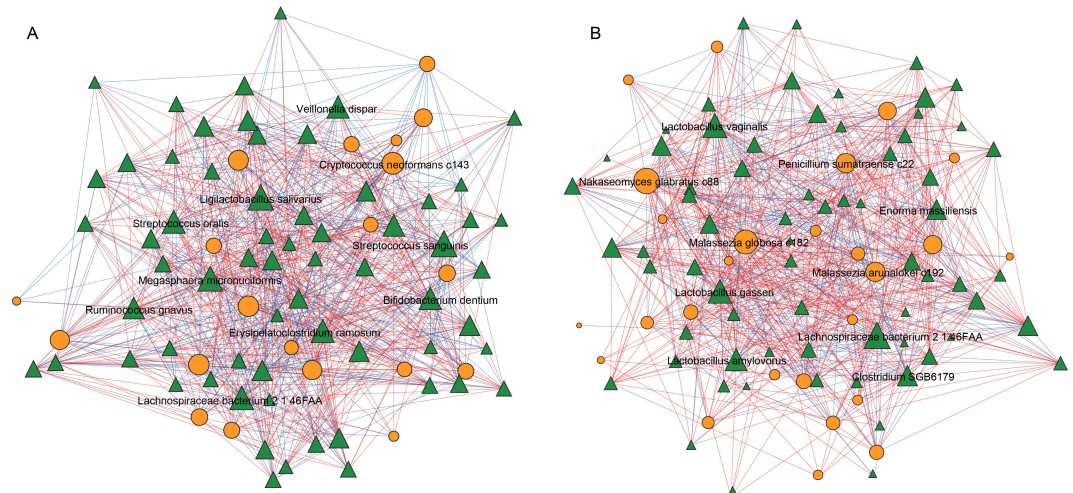

**FIG 4** Correlation analysis among gut bacteriome and mycobiome. The network showed correlations between groups of gut bacteria and fungi in ACVD patients (A) and the HC group (B), and labeled the species with the top 10 largest number of connections in the network.

mycobiome in ACVD patients. Consistent with previous findings on the gut bacteriome (10), the gut mycobiome in ACVD patients significantly differs from that of healthy controls. Meanwhile, we observed a notable increase in fungal richness among ACVD patients, aligning with a prior study on hypertension using the internal transcribed spacer sequencing method (22). Interestingly, an increase in gut fungal diversity has also been reported in patients with various immune system diseases, such as inflammatory bowel disease (23).

Regarding fungal taxa composition, Saccharomycotina, Pezizomycotina, and Basidiomycota were the predominant subphyla, collectively accounting for over 90% of total abundance across groups. *Penicillium* was identified as the predominant genus at the genus level, a finding consistent with research on hypertriglyceridemia. Research has demonstrated a close association between the proliferation of *Penicillium* and conditions such as hypertriglyceridemia (24) and obesity (25). Furthermore, differential analysis at the genus level reveals a significant increase in the abundance of *Penicillium* in patients with ACVD (FC >1.5). Given that both hypertriglyceridemia and obesity are risk factors for ACVD, it is reasonable to hypothesize that *Penicillium* may play a significant role in the development of ACVD.

Comparative analysis of fungal community composition revealed 25 species with significant differences between ACVD patients and healthy controls (FC >1.2, LDA >2,

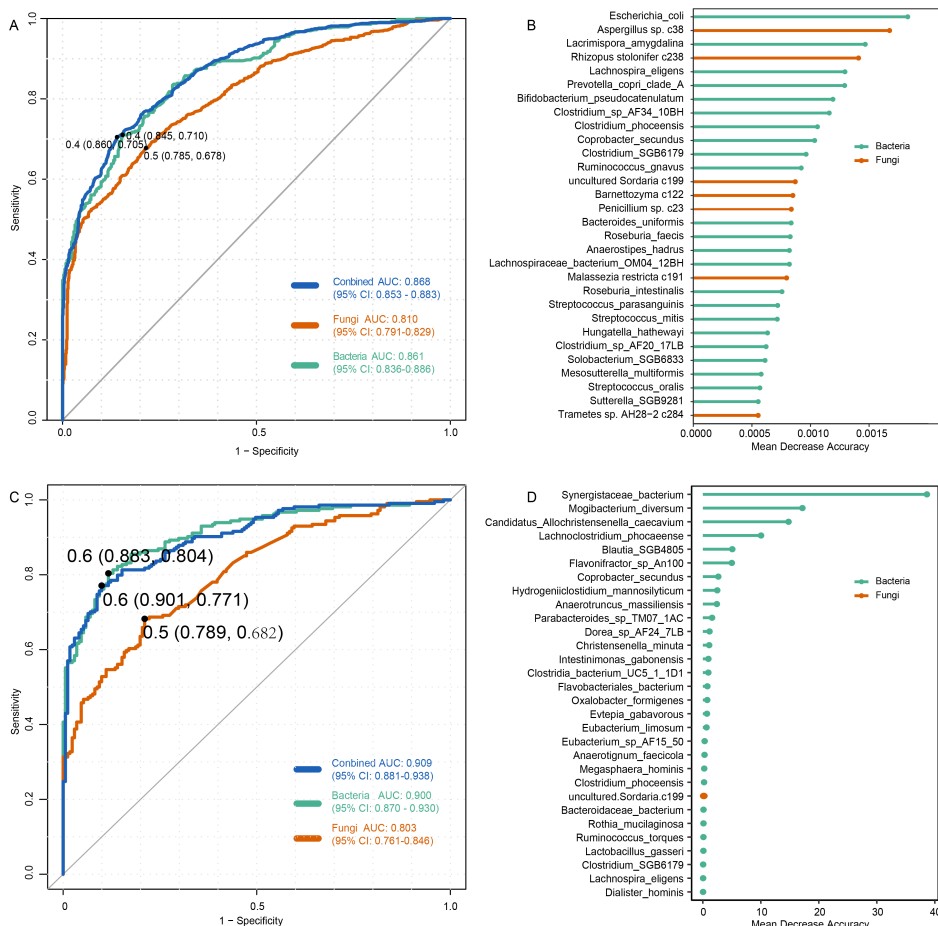

FIG 5 Classification of ACVD status by the abundances of gut bacteriome and mycobiome. (A) Receiver operating characteristic (ROC) analysis for classification of ACVD status using the gut bacterial and fungal signatures derived from a random forest model. (B) The 30 most discriminant signatures identified by the random forest model, with bar lengths indicating variable importance. (C) ROC analysis for classification of ACVD status using the gut bacterial and fungal signatures derived from a LASSO model. (D) The 30 most discriminant signatures identified by the LASSO model, with bar lengths indicating variable importance.

and relative abundance >0.01). Among them, 12 species were enriched in ACVD patients, including *Malassezia restricta* c191, *Exophiala spinifera* c165, *Exophiala dermatitidis* c164, *Candida albicans* c115, *Malassezia furfur* c28, and *Malassezia globosa* c182. Fungal infections caused by these pathogens have been reported in various diseases (26, 27), with the most notorious being *Candida albicans*. Studies have shown that *Candida albicans* accelerates atherosclerosis by activating intestinal hypoxia-inducible factor 2α signaling (27). Additionally, *Malassezia* spp. manifest multiple proinflammatory biological properties (28) and can promote the development of inflammatory-associated diseases, such as hypertension, Crohn's disease, and inflammatory bowel disease (22, 29, 30). *Exophiala* spp., though relatively uncommon, are important opportunistic pathogens causing subcutaneous or even fatal disseminated infections in both immunosuppressed and healthy individuals (31, 32), with various reports describing this genus as an etiologic agent of phaeohyphomycosis (33). Similarly, the two genera with the most significant differences were *Malassezia* and *Exophiala*. These genera may serve as important disease markers and potentially play a significant role in the development of ACVD. In contrast, only five species were enriched in the healthy controls. Some studies have shown that *Rhizopus stolonifer* significantly increases the abundance of cecal *Bifidobacterium* and *Lactobacillus* in mice (34), suggesting it may play an important role in maintaining gut homeostasis. However, the other four species did not display clear probiotic characteristics, which may indicate the presence of a unique dysbiosis profile in the gut microbiota of ACVD patients.

Fungi and bacteria co-exist and interact in the guts of humans and animals, either through mutualistic relationships or through competition. For example, in animal models of colitis, Enterobacteriaceae can cooperate with certain fungi, aiding their colonization and actively promoting inflammation (35). Investigating the differences in bacterial and fungal networks between ACVD patients and healthy individuals may help elucidate variations in microbial interactions or proinflammatory mechanisms. Through network analysis based on differential taxa, we identified microbial taxa that may be crucial for host biological functions. We identified four fungi that play significant roles in the gut microbiota network of ACVD patients: *Malassezia globosa* c182, *Penicillium sumatraense* c22, *Nakaseomyces glabratus* c88, and *Malassezia arunalokei* c192. Previous studies have shown that *Nakaseomyces glabratus* (*Candida glabrata*) is likely a commensal species in the human digestive tract, but systemic infections in immunocompromised patients can be fatal (36). *Malassezia globose*, a species of *Malassezia*, can induce proinflammatory cytokine IL-1β production and activate the NLRP3 inflammasome in phagocytes (37). Although the specific role of *Penicillium sumatraense* C22 in the gut remains unclear, the findings discussed earlier, which link the *Penicillium* genus to hypertriglyceridemia and obesity, along with its significantly elevated abundance in patients with ACVD, collectively suggest that *Penicillium sumatraense* C22 may play a role in the pathogenesis of ACVD. These findings suggest that fungi play a significant role in the development of ACVD, particularly the genus *Malassezia*. Therefore, further research on the specific roles of these fungal taxa in ACVD is necessary in the future.

We trained random forest and LASSO models based on gut fungal and bacterial characteristics for disease differentiation, achieving high predictive accuracy for distinguishing between healthy controls and patients. In both models, those built on bacterial features outperformed those based on fungal features, suggesting that bacteria may have a closer association with the progression of ACVD compared to fungi. Despite bacterial identification taxa demonstrating superior discriminative ability in the prediction model, certain fungi still played a significant role. These results indicate that the unique roles of these fungal taxa in ACVD warrant further investigation.

Although our study provides initial insights into the role of the gut mycobiome in ACVD, there are still some limitations that future research should address. Although we have employed appropriate statistical analysis methods and databases, there is still a certain degree of influence of diet and environment on the gut microbiome. The lack of longitudinal data limits the ability to monitor and analyze changes in the gut

mycobiome over time in ACVD patients. Therefore, future studies should incorporate regular sampling (e.g., every 3–6 months) from both ACVD patients and healthy controls. This approach would help identify specific fungal species associated with disease progression and provide insights into potential causal relationships, thereby improving understanding of the gut mycobiome's role in ACVD and informing targeted therapeutic strategies. Additionally, the current limitations in fungal detection tools, particularly regarding long-term culture, hinder a comprehensive understanding of pathogenic fungi (38). Developing more sensitive and efficient detection tools should be a priority in future research. Furthermore, studies should focus on investigating the impact of a single fungal species across multiple diseases. Understanding how one fungus operates in different disease contexts not only can reveal universal mechanisms of fungal pathogenicity but also can help identify new therapeutic targets (39). Such cross-disease research will enhance our understanding of the fungal ecosystem and its potential pathogenic roles in a range of diseases.

## Conclusions

Overall, we systematically characterized the gut mycobiome in patients with ACVD through metagenomic sequencing of their fecal samples. Compared to the gut mycobiome of healthy individuals, the gut mycobiome in ACVD patients has undergone significant alterations, particularly in the abundance of fungal species such as *Penicillium*, *Malassezia*, and *Exophiala*. Additionally, analysis of the gut fungal and bacterial microbiome networks in ACVD patients has revealed significant changes in interaction patterns and key taxa within the gut microbiota associated with ACVD. Our research will provide valuable insights for future mechanistic and clinical intervention studies. However, due to the lack of metabolic data or clinical information for these samples, we are currently unable to explore other potential risk factors that may contribute to gut fungal infections, such as trimethylamine N-oxide or diabetes. To comprehensively assess the impact of these factors, additional data sets are required for further validation.

## ACKNOWLEDGMENTS

This work was supported by the Harbin Medical University and by the Department of Laboratory Medicine, the First Affiliated Hospital of Harbin Medical University.

G. S., P. H., D. L. and G. X.contributed to bioinformatics and statistical analysis. R. G., S. F., and L. C. wrote the draft manuscript. S. L., Q. Y., and W. Y. reviewed and revised the manuscript. All authors contributed to the article and approved the submitted version.

## AUTHOR AFFILIATIONS

[1]Department of Laboratory Medicine, First Affiliated Hospital of Harbin Medical University, Harbin, China
[2]Department of Gastroenterology, Harbin First Hospital, Harbin, China
[3]Puensum Genetech Institute, Wuhan, China
[4]Department of Microbiology, College of Basic Medical Sciences, Dalian Medical University, Dalian, China
[5]Loudi Central Hospital, Loudi, China

## AUTHOR ORCIDs

Guangming Su  http://orcid.org/0000-0001-5774-3401
Guorui Xing  http://orcid.org/0009-0009-1309-8661
Shenghui Li  http://orcid.org/0000-0003-1071-8510
Qiulong Yan  http://orcid.org/0000-0002-2697-4562
Wei Yang  http://orcid.org/0000-0002-5140-8941

## AUTHOR CONTRIBUTIONS

Ping Huang, Data curation, Formal analysis, Investigation, Methodology.

## DATA AVAILABILITY

The raw sequencing reads for this study are available at the European Bioinformatics Institute database under accession number ERP023788. The data generated or analyzed during this study are included in the article/supplemental material.

## ADDITIONAL FILES

The following material is available online.

### Supplemental Material

**Table S1 (Spectrum02182-24-s0001.xlsx).** Sample information from 214 ACVD patients and 171 healthy volunteers.
**Table S2 (Spectrum02182-24-s0002.xlsx).** Detailed information of 307 nonredundant genomes.
**Table S3 (Spectrum02182-24-s0003.xlsx).** Detailed information of the 14 differential genus identified by Wilcox test.
**Table S4 (Spectrum02182-24-s0004.xlsx).** Detailed information of the 25 differential Fungal species identified by Wilcox test and LDA score (FC>1.2, LDA>2, Relative abundance > 0.01%).
**Table S5 (Spectrum02182-24-s0005.xlsx).** Detailed information of the 69 differential bacterial species identified by Wilcox test and LDA score (FC>1.2, LDA>2, Relative abundance > 0.01).
**Table S6 (Spectrum02182-24-s0006.xlsx).** Detailed information of the fungal and bacterial interaction network topological properties in ACVD patients and health control groups.

### Open Peer Review

**PEER REVIEW HISTORY (review-history.pdf).** An accounting of the reviewer comments and feedback.

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
