## [Reviewer comments · Microbiology Spectrum]

Microbiology Spectrum

Gut mycobiome alterations and network interactions with the bacteriome in patients with atherosclerotic cardiovascular disease

Guangming Su, Ping Huang, Dan Liu, Guorui Xing, Ruochun Guo, Shenghui Li, Shao Fan, Lin Cheng, Qiulong Yan, and Wei Yang

Corresponding Author(s): Wei Yang, First Affiliated Hospital of Harbin Medical University

Review Timeline:

Submission Date:	August 30, 2024
Editorial Decision:	September 23, 2024
Revision Received:	November 4, 2024
Editorial Decision:	November 9, 2024
Revision Received:	November 13, 2024
Accepted:	November 14, 2024

Editor: Benjamin Liu

Reviewer(s): Disclosure of reviewer identity is with reference to reviewer comments included in decision letter(s). The following individuals involved in review of your submission have agreed to reveal their identity: Mushtak T.S. Al-Ouqaili (Reviewer #3)

Transaction Report:

DOI: <https://doi.org/10.1128/spectrum.02182-24>

Re: Spectrum02182-24 (**Gut mycobiome alterations and network interactions with the bacteriome in patients with atherosclerotic cardiovascular disease**)

Dear Dr. Wei Yang:

Thank you for the privilege of reviewing your work. Below you will find my comments, instructions from the Spectrum editorial office, and the reviewer comments.

Editor's comments:

There are grammar and syntax issues that are labelled in the comments in the manuscript that will be returned to the authors. Please address them accordingly.

Revision Guidelines

Sincerely,
Benjamin Liu
Editor
Microbiology Spectrum

Reviewer #1 (Comments for the Author):

Based on the paper "Gut mycobiome alterations and network interactions with the bacteriome in patients with atherosclerotic

cardiovascular disease," here is a comprehensive review with revision requests, comments on strengths and limitations, and evaluations of key sections of the paper:

Revision Requests:

Clarify Methodological Details:

Provide more detail on the quality control steps for sequencing data (section 2.3). Specifically, explain the thresholds for the "mean complexity under 30%" criterion.

Expand the description of the iDIRECT approach used for network construction to clarify how indirect correlations were efficiently eliminated.

Improve Statistical Analysis Transparency:

Include confidence intervals for alpha diversity metrics (Shannon and richness indices) to better understand variability within groups.

When using the random forest classifier, provide information on how class imbalance was handled, as the study involved different proportions of ACVD patients and healthy controls.

Address Limitations Explicitly:

The absence of longitudinal data is briefly mentioned. It would be valuable to expand on this limitation and suggest how future studies could implement longitudinal designs to observe dynamic changes in the gut mycobiome.

Acknowledge that confounding factors like diet, medication, or lifestyle were not controlled in this dataset and might affect the gut mycobiome.

Results Interpretation and Biological Relevance:

The paper discusses the increase of certain fungal taxa in ACVD patients, particularly *Candida albicans* and *Malassezia* species. Offer a mechanistic explanation or hypothesis as to why these specific taxa might contribute to the pathogenesis of atherosclerosis beyond simple proinflammatory activity.

References and Literature Review:

While there are sufficient references to recent studies, the paper would benefit from additional comparison to other metagenomic studies on the gut mycobiome, especially in cardiovascular diseases, to provide context for the findings.

Detailed Comments:

Introduction:

Strengths:

The introduction clearly establishes the knowledge gap by emphasizing the underexplored role of the gut mycobiome in ACVD compared to the better-studied bacteriome.

It logically builds a case for the relevance of studying fungal communities, citing examples of *Candida* and *Malassezia*'s role in inflammatory responses, which is well-supported by literature.

Suggestions:

Although the introduction is concise, more background information on previous findings regarding the gut mycobiome in metabolic disorders could further justify the study's relevance.

Materials and Methods:

Strengths:

The methods are robust, particularly the use of a tailored fungal genome database to accurately profile the gut mycobiome. Employing the PERMANOVA and Bray-Curtis distance-based PCoA for community composition analysis is appropriate for understanding microbial differences.

Limitations:

The study relies on a publicly available dataset, but the paper does not describe how external factors like sample handling or storage might have influenced the results. A section on sample integrity could improve methodological transparency.

The criteria for species detection thresholds and statistical filtering ($LDA > 2$, $FC > 1.2$) are not thoroughly explained.

Results:

Strengths:

The results section effectively uses visualizations (PCoA plots, sunburst diagrams) to convey fungal community differences between ACVD patients and healthy controls.

The identification of key taxa, such as *Malassezia* and *Candida*, provides concrete findings that could guide future mechanistic studies.

Limitations:

The network analysis showing altered microbial interactions in ACVD patients is intriguing but could be better contextualized. For

instance, what functional pathways might be affected by the shift towards fungal dominance in ACVD?

Suggestions:

Include a table summarizing the functional or pathogenic roles of the top fungal taxa identified, to help readers better understand the biological relevance of these findings.

Discussion and Conclusion:

Strengths:

The discussion ties the findings back to the growing body of research on microbiota-host interactions in cardiovascular disease, reinforcing the importance of gut fungi.

The authors correctly identify limitations, such as the absence of metabolic or clinical data in this study, which would be needed to explore causal links between gut fungi and ACVD.

Suggestions:

A clearer discussion of how these findings could translate into clinical practice or therapy (e.g., fungal-targeted probiotics) would enhance the paper's impact.

Provide more speculative insight into how fungi interact with bacteria in this altered network and whether the shift in fungal populations is a cause or consequence of ACVD.

Tables and Figures:

Strengths:

Figures are well-designed, particularly the sunburst diagram (Fig. 1) and PCoA plots (Fig. 2), which are intuitive for understanding the results.

Suggestions:

Consider adding annotations to the figures showing the statistical significance levels directly in the visualizations to make the results clearer at a glance.

Improve clarity in table legends by providing more context on how values (e.g., relative abundance, fold change) were derived.

Reviewer #3 (Comments for the Author):

Dear authors you are doing well and that looks great work. A few issues, however, need to be addressed;

In line 25 Could the authors explain the specifics of the shotgun metagenomic sequencing method employed? Were any particular factors taken into account to guarantee precise fungal identification?

In line 27 How did you handle the difficulties associated with identifying temporary and commensal fungal species in the gut microbiome?

In line 29 What statistical methods were used to compare the fungal communities between ACVD patients and healthy controls?

In line The introduction, which mostly focuses on bacterial communities, Highlights the role of the gut microbiota on ACVD.

Could you explain why, in this specific scenario, the gut mycobiome has received very little research?

In line 63 To enhance the introduction section add the following updated reference: Al-Moghira Khairi Al-Qaysi; Safaa Abed Latef Al-Meani; Mushtak T.S. Al-Ouqaili. (2020). The Effect of Dual-Species Biofilms, Monosaccharide and D-Amino Acids on Pseudomonas Biofilm. Indian Journal of Forensic Medicine & Toxicology, 15(1), 2177-2192.

<https://doi.org/10.37506/ijfmt.v15i1.13728>

In line 70 To enhance the introduction section add the following update reference: Jomehzadeha, N, Javaherizadehd, H, Amin M, Saki M, Al-Ouqaili MTS, Hamidic H, Seyedmahmoudic M, Gorjiana Z. (2021). Isolation and identification of potentially probiotic Lactobacillus species from faeces of infants in southwest Iran. International Journal of Infectious Diseases 96: 524-530.

In line 73 Why did you undertake a thorough research that included bacteria, fungi, and other microorganisms, instead of concentrating exclusively on fungal communities, given the intricate interactions between these two groups of microbes in the gut?

In line 76 Malassezia and Candida can alter immunological responses, according to the investigators. Are these fungi thought to affect the aetiology of ACVD through any particular mechanisms?

In line 92 How were the subjects selected for the study? Were there any inclusion or exclusion criteria for the patients and healthy controls?

Was the study prospective or retrospective?

In line 93 Any preprocessing steps applied to the metagenomic data before analysis, such as quality filtering, adapter removal, and trimming.

In line 132 Give a thorough explanation of the normalization procedure that was utilized to determine the relative abundances. Talk about the genome size that was taken into account and if any further normalizing methods were used.

In line 134 Before determining relative abundances, describe the normalization process used for the readings (e.g., TPM, RPKM, or other normalizing approaches).

In line 185 Detailed information on the characteristics of the ACVD (atherosclerotic cardiovascular disease) patients and healthy controls (e.g., age, gender, lifestyle factors, medication use).

In line 187 The authors report mild but significant separation between groups in PCoA. Is this separation biologically meaningful given the variance explained (18.4% and 10.4%)?

Does Table S3 provide enough information on the fold change and relative abundance of all significant genera? Is there a

reason for choosing a cutoff of FC > 1.2 and relative abundance >0.01?

In line 218 Are there known associations between the enriched genera (e.g., *Candida*, *Malassezia*, *Exophiala*) and ACVD pathophysiology? Could their enrichment be related to factors other than ACVD, such as diet, environment, or medication use?

In line 257 Why was a random forest classifier chosen for this analysis? Were other machine learning models (e.g., support vector machines, logistic regression) considered or tested?

How was the random forest model tuned (e.g., number of trees, depth of trees) to optimize performance?

In line 287 In what way does the discovery of elevated fungal diversity in patients with ACVD correspond with the current understanding of microbial dysbiosis in cardiovascular disorders? Exist conflicting results in the literature?

In line 295 Given the potential role of certain fungi in ACVD pathogenesis, are there any known antifungal treatments or dietary interventions that could be explored for therapeutic benefit?

In line 298 To enhance the discussion section add the following reference: Al-Ouqaili, M.T.S., Musleh, M.H., Al-Kubaisi, S.M.A. Depending on HPLC and PCR, detection of aflatoxin B1 extracted from *Aspergillus flavus* strains and its cytotoxic effect on AFB1 treated-hematopoietic stem cells obtained from human umbilical cord. *Asian Journal of Pharmaceutics*, Volume 12, Issue 3, July-September 2018, Pages S1048-S1054.

In line 351 Outline specific future research avenues that could build on this study, such as investigating the functional roles of identified fungal taxa in ACVD, exploring the relationship between the mycobiome and metabolic risk factors (like TMAO), or conducting longitudinal studies to assess how gut mycobiome changes correlate with disease progression.

**Gut mycobiome alterations and network interactions with the**
**bacteriome in patients with atherosclerotic cardiovascular disease**

Guangming Su^{1#}, Dan Liu^{1#}, Guorui Xing^{2#}, Ruochun Guo², Shenghui Li², Shao Fan³,
Lin Cheng³, and Qiulong Yan^{3*}, Wei Yang^{1*}

¹ Department of Laboratory Diagnostics, First Affiliated Hospital of Harbin Medical
University, Harbin, China

² Puensum Genetech Institute, Wuhan, China

³ Department of Microbiology, College of Basic Medical Sciences, Dalian Medical
University, Dalian, China

* Corresponding author.

# Contributed equally.

Address correspondence to Wei Yang, yangwei6295@163.com

Address correspondence to Qiulong Yan, Qiulongy1988@163.com

Guangming Su, Dan Liu, and Guorui Xing contributed equally to this article. Author
order was determined by the corresponding author after negotiation.

The authors declare no conflict of interest.

**Abstract**

[revised manuscript text omitted]

The newly established iDIRECT approach was used to construct fungal and bacterial
co-occurrence network for two groups, which efficiently eliminated indirect correlations
and quantitatively inferred direct dependencies in a network (18). To minimize false-
positive, only species detected in more than 10% biological replicates for each group
were included. We focused on the interactions among taxa involved in bacteria and fungi.
Networks were visualized by Cytoscape v3.8.2 (19).

The random forest classifier based on the gut mycobiome was built using the
'*randomForest*' function followed by 5 times five-fold cross-validations, and their
performances were evaluated based on area under the receiver operator characteristic
curve (AUC) that was calculated by the '*roc*' function. The importance ordering of
markers was obtained via the '*importance*' function.

**Results**

**Sample information and fungal database**

To characterize the gut mycobiome community in patients with ACVD, we analyzed the
metagenomic sequencing data set from fecal samples of 214 patients and 171 healthy
controls. To accurately define the mycobiome's composition, we used a tailored fungal
database utilizing strict filtering criteria (see Materials and methods for details). This
refined database includes 307 distinct reference species, categorized based on a 95%
average nucleotide identity (ANI) criterion, derived from an assortment of 1,490
genomes linked to the human microbiome (Table S2). Subsequently, high-quality reads
from each sample were aligned to the genomes of these 307 distinct species present in
our database, facilitating the creation of in-depth mycobiome profiles. Furthermore, our
examination revealed the presence of 178 high-level taxa across the samples, which

comprised 84 genera, 49 families, 27 orders, 14 classes, and 5 subphyla (Fig. 1). To
determine the effects of gender, age, and BMI on the gut mycobiome, we utilized
PERMANOVA to compare the impacts of these three indicators. The findings revealed
that none of the factors—gender (Adonis = 0.0043, $p=0.169$), age (Adonis = 0.0035,
$p=0.171$), or BMI (Adonis = 0.0037, $p=0.129$)—significantly affected the structure of the
gut fungal community.

**Altered gut mycobiome structure in ACVD patients**

We compared the gut mycobiome composition between healthy controls and ACVD
patients using PCoA and PERMANOVA. Bray-Curtis distance-based PCoA of species-
level composition revealed that PCoA1 and PCoA2 accounted for 18.4% and 10.4% of
the total variation, respectively (Fig. 2a). ACVD patients showed a mild but statistically
significant separation from healthy controls along PCoA1 (Wilcoxon rank-sum test
$p<0.05$). PERMANOVA confirmed significant differences in the gut mycobiome between
the groups (Adonis = 0.014, $p<0.001$). Alpha diversity, assessed using richness and
Shannon indices, indicated higher mycobiome richness in ACVD patients compared to
healthy controls (Wilcoxon rank-sum test $p<0.001$, Fig. 2b), while Shannon's index
showed no significant difference between the groups.

In terms of the fungal taxa, the gut mycobiome of all subjects was usually dominated
by Saccharomycotina, followed by Pezizomycotina, Basidiomycota, and Mucoromycota
(Fig. 2c). At the genus level, *Penicilium* was the first most abundant genus, while other
common genera, such as *Pichia*, *Barnettozyma*, and *Barnettozyma*, had relatively high
abundances in both groups (Fig. 2d). According to the comparison analysis, a total of 14
genera exhibited significant differences (FC>1.2, Relative abundance >0.01,  Table S3)
between ACVD patients and healthy controls (Fig. 2e). Among these genera, two stood
out with the most prominent differences (FC>10), namely *Exophiala* (Wilcoxon rank-
sum test, adjusted p < 0.001) and *Malassezia* (adjusted p < 0.001).

**Gut fungal signatures associated with ACVD**

Based on the significant differences observed at the genus level, we further investigated
the differential taxa at the species level, aiming to identify potential biomarkers
associated with ACVD. To achieve this, we applied stringent filters, considering FC>1.2,
LDA>2, and an average relative abundance greater than 0.01. The analysis revealed a
total of 25 significantly different fungal taxa, spanning across 7 classes and 13 genera
(Fig. 3; Table S4). At the class levels, 5 fungal taxa were significantly enriched in ACVD
patients, including the classes Saccharomycetes, Sordariomycetes, Malasseziomycetes,

Tremellomycetes and Agaricomycetes. At the genus level, 8 fungal populations, including
*Candida*, *Malassezia*, *Exophiala*, *Nakaseomyces*, *Bamettozyma*, *Microascales*,
*Cryptococcus* and *Irpex*, were significantly enriched in ACVD patients, whereas *Rhizopus*
and *Trichophyton* were significantly enriched in healthy controls. At the species level, 20
fungal species were enriched in ACVD patients, while 5 species were enriched in healthy
controls. Particularly noteworthy was the significant increase in the gut fungus *Candida*
*albicans* among ACVD patients compared to healthy controls. In addition, various
opportunistic pathogens, including *Aspergillus* sp., *Exophiala spinifera*, *Malassezia*
*restricta*, *Candida albicans*, *Mucor* sp., and *Malassezia furfur*, were found to be enriched
in ACVD patients.

**Comparative analysis of gut microbiome co-occurrence networks**

Due to the coexistence of multiple microorganisms in the gut, including bacteria and
fungi, which actively collaborate and collectively form the gut microbiota network, it is
crucial to further investigate the interplay among the gut microbiome and mycobiome. To
be more precise, we applied uniform criteria to identify bacterial species that exhibited
significant differences between individuals with ACVD and those who were healthy
(Table S5). These differentially abundant taxa were subsequently utilized to construct a

[revised manuscript text omitted]

**Acknowledgments**

The authors greatly appreciate all the patients involved in the study. All authors read and
gave final approval of the version to be submitted.

**Data Availability Statement**

The raw sequencing reads for this study are available at the European Bioinformatics
Institute (EBI) database under accession number ERP023788. And the data generated or
analyzed during this study are included in the article/supplementary material.

**Conflict of interest**

The authors have no conflicts of interest to report.

**References**

- 1. Frostegård JJBm. 2013. Immunity, atherosclerosis and cardiovascular disease.
11:1-13.
- 2. Chaldakov GN, Fiore M, Ghenev PI, Stankulov IS, Aloe LJIMJ. 2000. Adipose
Tissue. 7:43-49.
- 3. Lusis AJ. 2000. Atherosclerosis. Nature 407:233-41.
- 4. Salekeen R, Haider AN, Akhter F, Billah MM, Islam ME, Islam KMDJIJoCCR,
Prevention. 2022. Lipid oxidation in pathophysiology of atherosclerosis: Current
understanding and therapeutic strategies. 14:200143.

- 5. Yoo JY, Sniffen S, McGill Percy KC, Pallaval VB, Chidipi BJM. 2022. Gut
dysbiosis and immune system in atherosclerotic cardiovascular disease (ACVD).
10:108.
- 6. Boutari C, Hill MA, Procaccini C, Matarese G, Mantzoros CSJM. 2023. The key
role of inflammation in the pathogenesis and management of obesity and CVD, p
155627. Elsevier.
- 7. Bäckhed F, Ding H, Wang T, Hooper LV, Koh GY, Nagy A, Semenkovich CF,
Gordon JIJPotnaos. 2004. The gut microbiota as an environmental factor that
regulates fat storage. 101:15718-15723.
- 8. Buford TWJM. 2017. (Dis) Trust your gut: the gut microbiome in age-related
inflammation, health, and disease. 5:1-11.
- 9. Jie Z, Xia H, Zhong S-L, Feng Q, Li S, Liang S, Zhong H, Liu Z, Gao Y, Zhao
HJNc. 2017. The gut microbiome in atherosclerotic cardiovascular disease.
8:845.
- 10. Li XV, Leonardi I, Iliev IDJI. 2019. Gut mycobiota in immunity and
inflammatory disease. 50:1365-1379.

- 11. Richard ML, Sokol HJ*N*RG, Hepatology. 2019. The gut mycobiota: insights into
analysis, environmental interactions and role in gastrointestinal diseases. 16:331-
345.
- 12. Yan Q, Li S, Yan Q, Huo X, Wang C, Wang X, Sun Y, Zhao W, Yu Z, Zhang YJC.
2024. A genomic compendium of cultivated human gut fungi characterizes the gut
mycobiome and its relevance to common diseases. 187:2969-2989. e24.
- 13. Almeida A, Nayfach S, Boland M, Strozzi F, Beracochea M, Shi ZJ, Pollard KS,
Sakharova E, Parks DH, Hugenholtz PJ*N*b. 2021. A unified catalog of 204,938
reference genomes from the human gut microbiome. 39:105-114.
- 14. Quast C, Pruesse E, Yilmaz P, Gerken J, Schweer T, Yarza P, Peplies J, Glöckner
FOJ*N*ar. 2012. The SILVA ribosomal RNA gene database project: improved data
processing and web-based tools. 41:D590-D596.
- 15. Dixon PJ*J*ovs. 2003. VEGAN, a package of R functions for community ecology.
14:927-930.
- 16. Asnicar F, Thomas AM, Beghini F, Mengoni C, Manara S, Manghi P, Zhu Q,
Bolzan M, Cumbo F, May UJ*N*c. 2020. Precise phylogenetic analysis of microbial
isolates and genomes from metagenomes using PhyloPhlAn 3.0. 11:2500.

- 17. Letunic I, Bork PJ*Nar*. 2019. Interactive Tree Of Life (iTOL) v4: recent updates
and new developments. 47:W256-W259.
- 18. Xiao N, Zhou A, Kempfer ML, Zhou BY, Shi ZJ, Yuan M, Guo X, Wu L, Ning D,
Van Nostrand JJPotNAoS. 2022. Disentangling direct from indirect relationships
in association networks. 119:e2109995119.
- 19. Su G, Morris JH, Demchak B, Bader GD*JCpib*. 2014. Biological network
exploration with Cytoscape 3. 47:8.13. 1-8.13. 24.
- 20. Zou Y, Ge A, Lydia B, Huang C, Wang Q, Yu YJ*FiI*. 2022. Gut mycobiome
dysbiosis contributes to the development of hypertension and its response to
immunoglobulin light chains. 13:1089295.
- 21. Sokol H, Leducq V, Aschard H, Pham H-P, Jegou S, Landman C, Cohen D,
Liguori G, Bourrier A, Nion-Larmurier IJG. 2017. Fungal microbiota dysbiosis in
IBD. 66:1039-1048.
- 22. Ahmad HF, Castro Mejia JL, Krych L, Khakimov B, Kot W, Bechshøft RL,
Reitelseder S, Højfeldt GW, Engelsen SB, Holm LJB. 2020. Gut Mycobiome
dysbiosis is linked to hypertriglyceridemia among home Dwelling elderly
Danes.2020.04. 16.044693.

- 23. Mar Rodríguez M, Pérez D, Javier Chaves F, Esteve E, Marin-Garcia P, Xifra G,
Vendrell J, Jové M, Pamplona R, Ricart WJSr. 2015. Obesity changes the human
gut mycobiome. 5:14600.
- 24. Neoh CF, Chen SC, Crowe A, Hamilton K, Nguyen QA, Marriott D, Trubiano JA,
Spelman T, Kong DC, Slavin MA. Invasive scedosporium and Lomentospora
prolificans infections in Australia: a multicenter retrospective cohort study, p
ofad059. In (ed), Oxford University Press US,
- 25. Wang X, Zhou S, Hu X, Ye C, Nie Q, Wang K, Yan S, Lin J, Xu F, Li MJCH,
Microbe. 2024. Candida albicans accelerates atherosclerosis by activating
intestinal hypoxia-inducible factor2 α signaling.
- 26. Roesner L, Ernst M, Chen W, Begemann G, Kienlin P, Raulf M, Lepenies B,
Werfel TJSr. 2019. Human thioredoxin, a damage-associated molecular pattern
and Malassezia-crossreactive autoallergen, modulates immune responses via the
C-type lectin receptors Dectin-1 and Dectin-2. 9:11210.
- 27. Limon JJ, Tang J, Li D, Wolf AJ, Michelsen KS, Funari V, Gargus M, Nguyen C,
Sharma P, Maymi VIJCh, microbe. 2019. Malassezia is associated with Crohn's
disease and exacerbates colitis in mouse models. 25:377-388. e6.

- 28. Yang Q, Ouyang J, Pi D, Feng L, Yang JF*et al.* 2022. *Malassezia* in inflammatory
bowel disease: Accomplice of evoking tumorigenesis. *13:846469*.
- 29. Zeng J, Sutton D, Fothergill A, Rinaldi M, Harrak M, de Hoog GJ*et al.* 2007.
Spectrum of clinically relevant *Exophiala* species in the United States. *45:3713-*
*3720*.
- 30. Kotylo PK, Israel KS, Cohen JS, Bartlett MS*et al.* 1989. Subcutaneous
phaeohyphomycosis of the finger caused by *Exophiala spinifera*. *91:624-627*.
- 31. Haridasan S, Parameswaran S, Bheemanathi SH, Chandrasekhar L, Suseela BB,
Singh R, Rabindranath J, Padhi RK, Sampath E, Dubey AK*et al.* 2017.
Subcutaneous phaeohyphomycosis in kidney transplant recipients: a series of
seven cases. *19:e12788*.
- 32. Yang Y, Kameda T, Aoki H, Nirmagustina DE, Iwamoto A, Kato N, Yanaka N,
Okazaki Y, Kumrungsee T*et al.* 2018. The effects of tempe fermented with
*Rhizopus microsporus*, *Rhizopus oryzae*, or *Rhizopus stolonifer* on the colonic
luminal environment in rats. *49:162-167*.
- 33. Sovran B, Planchais J, Jegou S, Straube M, Lamas B, Natividad JM, Agus A,
Dupraz L, Glodt J, Da Costa GJM. 2018. Enterobacteriaceae are essential for the
modulation of colitis severity by fungi. *6:1-16*.

- 34. Bolotin - Fukuhara M, Fairhead CJY. 2014. *Candida glabrata*: a deadly
companion? 31:279-288.
- 35. Wolf AJ, Limon JJ, Nguyen C, Prince A, Castro A, Underhill DMJJoLB. 2021.
*Malassezia* spp. induce inflammatory cytokines and activate NLRP3
inflammasomes in phagocytes. 109:161-172.

Figure legends

**Fig. 1** Sunburst diagram of taxonomic hierarchy for 307 gut fungal species and 178 high-
level taxa

**Fig. 2** Comparison of gut mycobiome diversity and structure between ACVD patients and
healthy controls. (A) PCoA based on Bray–Curtis distance of the fungal profiles at
the species level. The plot displayed the distribution of samples along PCoA1 and
PCoA2, with ellipsoids indicating the 95% confidence interval for each group.
The upper and right boxplots displayed the sample scores in PCoA1 and PCoA2.
(B) Comparison of alpha diversity indexes between ACVD patients and healthy
controls. The p-value was determined by the Wilcoxon rank-sum test. (C) Pie
chart showing the composition of fungal subphyla in each group. The
percentages represented the average relative abundance of each subphylum. (D)
Distribution of the top 10 abundant genera across all samples. (E) Boxplots
showing the relative abundances of the genera *Exophiala* and *Malassezia* in each

group (FC>10, Relative abundance>0.01). Statistical significance was determined
using the Wilcoxon rank-sum test with Benjamini and Hochberg adjustment.

**Fig. 3** The taxonomic tree of all differentially enriched species. Each tree-like branch
with a different color represents a single phylum. The bar chart outside the fan
chart describes the class level classification of a particular species. Species that
were enriched in ACVD patients are shown as red triangles, and depleted species
are shown as green triangles

**Fig.4** Correlation analysis among gut bacteriome and mycobiome. The network showed
correlations between groups of gut bacteria and fungi in ACVD patients (A) and
HC group (B), and labeled the species with the top ten largest number of
connections in the network.

**Fig. 5** Classification of ACVD status by the abundances of gut bacteriome and
mycobiom. (A) ROC analysis for classification of ACVD status using the gut
bacterial and fungal signatures. (B) The 30 most discriminant signatures in the
model classifying ACVD patients and healthy controls. The bar lengths indicate

the importance of the variable. (C) Exploring the classification performance for
different numbers of bacterial/fungal signatures ordered in importance. Nodes
show the average AUC of models with 10 repetitions under a specified number of
signatures, and the error bars show the standard deviations.

Responses to the Reviewers' comments

Reviewer #1 (Comments for the Author):

Based on the paper "Gut mycobiome alterations and network interactions with the bacteriome in patients with atherosclerotic cardiovascular disease," here is a comprehensive review with revision requests, comments on strengths and limitations, and evaluations of key sections of the paper:

Response: Thank you for acknowledging the significance of our study and for your thorough review of our manuscript. We have carefully considered your suggestions and have made the necessary revisions and additions. We hope our responses meet your expectations.

1. Provide more detail on the quality control steps for sequencing data (section 2.3). Specifically, explain the thresholds for the "mean complexity under 30%" criterion.

Response: Thank you for examining this aspect. Regarding the threshold for the "mean complexity under 30%" criterion, we selected this specific cutoff to effectively filter out low-complexity reads that could potentially introduce biases or reduce the overall quality of our metagenomic data. For instance, a read predominantly consisting of a simple repetitive sequence like "ATATATATAT" or a homopolymeric stretch such as "AAAAAAA" would have a low complexity score. These sequences are less useful for distinguishing between different microbial taxa or for accurate genome assembly. By setting a mean complexity threshold at 30%, we ensure that such low-complexity reads are effectively removed, thereby enhancing the reliability of our metagenomic analyses.

To ensure data quality, we utilized fastp v0.20 to process each metagenomic sample. The raw reads underwent several quality control steps, including the trimming of polyG tails and the removal of low-quality reads based on the following criteria: (1) reads shorter than 90bp; (2) reads with a mean Phred quality score below 20; (3) reads with more than 30% of bases having a Phred

quality score below 20; (4) reads with a mean complexity under 30%—such as repetitive sequences like "ATATATATAT" or homopolymeric stretches like "AAAAAAA," which have low complexity scores; and (5) unpaired-end reads.

2. Expand the description of the iDIRECT approach used for network construction to clarify how indirect correlations were efficiently eliminated

Response: To investigate the differences in interaction patterns between fungi and bacteria in the gut of healthy individuals and patients with atherosclerotic cardiovascular disease (ACVD), we employed network methods based on Random Matrix Theory (RMT) to construct two co-occurrence networks using the platform available at <http://ieg2.ou.edu/MENA>. The RMT method ensures that the association strength adheres to a Poisson distribution under natural conditions. Additionally, we used iDIRECT (Inference of Direct and Indirect Relationships with Effective Copula-Based Transitivity) to eliminate potential spurious indirect connections in the original network, including pathological connections, self-loops, and overly strong interactions within the ecological network. The Cytoscape v3.8.2 software were used to visualize the networks.

3. Include confidence intervals for alpha diversity metrics (Shannon and richness indices) to better understand variability within groups.

Response: Thank you for your suggestion! We have now included confidence intervals for the Shannon and richness indices in the figures to better understand the variability within the groups.

4. When using the random forest classifier, provide information on how class imbalance was handled, as the study involved different proportions of ACVD patients and healthy controls.

Response: To address class imbalance due to differing proportions of ACVD patients and healthy controls, we used 5-fold cross-validation by allocating healthy and patient samples in each fold as one-tenth of their

respective total sample sizes, ensuring the class distribution remained consistent within each fold.

5. The absence of longitudinal data is briefly mentioned. It would be valuable to expand on this limitation and suggest how future studies could implement longitudinal designs to observe dynamic changes in the gut mycobiome.

Response: Additionally, our study has some limitations, and future research should aim to address them. For example, the lack of longitudinal data limits our ability to monitor and analyze changes in the gut mycobiome over time in patients with ACVD. Future studies should incorporate regular sampling (e.g., every 3-6 months) from both ACVD patients and healthy controls. This approach would help identify specific fungal species associated with disease progression and provide insights into potential causal relationships. Such findings could enhance understanding of the gut mycobiome's role in ACVD and inform targeted therapeutic strategies.

6. Acknowledge that confounding factors like diet, medication, or lifestyle were not controlled in this dataset and might affect the gut mycobiome.

Response: We acknowledge that diet and environment have certain impacts on the gut microbiome, but these impacts are filled with randomness. In our study design, we have made efforts to control these variables as much as possible and used appropriate statistical analysis methods to minimize their interference with the results. This point has been added to the manuscript in lines 407-409. All patients were ethnic Han Chinese with no known consanguinity, aged 40–80 years old. The exclusion criteria included ongoing infectious diseases, cancer, renal, or hepatic failure, peripheral neuropathy, stroke, as well as use of antibiotics within 1 month of sample collection. All the healthy control individuals enrolled were free of clinically evident ACVD symptoms at the time of the medical examination. Demographic data and cardiovascular risk factors were collected by a questionnaire. Individuals with

peripheral artery disease, known coronary artery disease or myocardial infarction, cardiomyopathy, renal failure, peripheral neuropathy, systemic disease, and stroke were excluded. Additionally, none of the ACVD patients had received steroids or antibiotics within the preceding three months.

7.The paper discusses the increase of certain fungal taxa in ACVD patients, particularly *Candida albicans* and *Malassezia* species. Offer a mechanistic explanation or hypothesis as to why these specific taxa might contribute to the pathogenesis of atherosclerosis beyond simple proinflammatory activity.

Response: Thank you for your insightful comments. The potential mechanisms by which these fungi might contribute to atherosclerosis include the activation of intestinal HIF-2 α signaling by formyl-methionine from *Candida albicans*, leading to increased ceramide synthesis and accelerated atherosclerosis, and the activation of immune responses via the CARD9 pathway by *Malassezia restricta*, which could enhance systemic inflammation and promote atherosclerotic plaque formation. However, these mechanisms require further investigation to clarify their roles in pathogenesis. Over-speculation may impact future research, so we recommend cautious interpretation while encouraging further studies to explore these pathways.

8.Although the introduction is concise, more background information on previous findings regarding the gut mycobiome in metabolic disorders could further justify the study's relevance.

Response: Thank you for your valuable feedback. We have revised the introduction section according to your suggestion. Specifically, we have added more background information on previous findings regarding the gut mycobiome in metabolic disorders in lines 72-86. These additions include discoveries from prior research and further elaboration on the significance of our study. We hope these revisions better highlight the relevance of our research and adequately address your review comments. Once again, thank

you for your thorough review and constructive feedback.

9.The study relies on a publicly available dataset, but the paper does not describe how external factors like sample handling or storage might have influenced the results. A section on sample integrity could improve methodological transparency.

Response: Thank you for your feedback. We have incorporated additional details regarding the DNA extraction and DNA library construction in the Methods section, specifically in lines 116-126. We hope these additions enhance the transparency of our study and meet your expectations for the Methods section.

10.The criteria for species detection thresholds and statistical filtering ($LDA > 2$, $FC > 1.2$) are not thoroughly explained.

Response: LDA (Linear Discriminant Analysis) values are used to measure the degree of separation between different groups based on specific classification variables. By setting an LDA threshold ($LDA > 2$), researchers can filter out noise and non-significant differences, thereby enhancing the reliability of the results. Additionally, Fold Change (FC) is used to assess changes in the abundance of a particular species under different conditions or groups. An FC threshold ($FC > 1.2$) helps identify biologically meaningful differences, avoiding misleading conclusions from minor variations. Combining these two standards ($LDA > 2$, $FC > 1.2$) allows for the accurate identification of statistically and biologically significant microbial differences, ensuring the credibility and scientific validity of the study results. In response to your comment, we have added a detailed explanation in lines 186-191. We appreciate your valuable feedback, which helps us improve the clarity and thoroughness of our study.

11.The network analysis showing altered microbial interactions in ACVD

patients is intriguing but could be better contextualized. For instance, what functional pathways might be affected by the shift towards fungal dominance in ACVD?

Response: Thanks for the reviewer for this question. In patients with ACVD, fungi within the gut microbiome play a significant role and may impact various functional pathways related to the disease. Firstly, *Malassezia globosa* can exacerbate inflammatory responses by inducing the production of the pro-inflammatory cytokine IL-1 β and activating the NLRP3 inflammasome. Secondly, *Nakaseomyces glabratus* (*Candida glabrata*) may influence immune regulation, as it can cause systemic infections in immunocompromised individuals. Additionally, while the specific role of *Penicillium sumatraense* C22 in the gut is not well-defined, its association with hypertriglyceridemia and obesity suggests it may be involved in metabolic pathways contributing to ACVD. These findings indicate that the role of fungi in ACVD patients warrants further investigation.

12. Provide more speculative insight into how fungi interact with bacteria in this altered network and whether the shift in fungal populations is a cause or consequence of ACVD.

Response: Thank you for raising this insightful question. Fungi and bacteria coexist in microbial communities, and they may engage in competition, symbiosis, or cooperation. For example, certain fungi might inhibit bacterial growth by secreting antimicrobial substances, while others may collaborate with bacteria to form biofilms, offering mutual protection. Changes in these interactions could influence the host's immune response and inflammation, thereby affecting disease progression.

As for whether the shift in fungal populations is a cause or consequence of ACVD, it remains unclear. Fungi might directly influence the host's immune system or metabolic pathways, promoting the development of ACVD. Conversely, ACVD itself might alter the host environment and immune status,

leading to changes in the fungal community. Future research, including longitudinal and mechanistic studies, will be essential to further clarify this causal relationship.

13. Include a table summarizing the functional or pathogenic roles of the top fungal taxa identified, to help readers better understand the biological relevance of these findings.

Response: Thank you to the reviewer for the valuable suggestion. In future research, we will further organize and present these results.

14. Consider adding annotations to the figures showing the statistical significance levels directly in the visualizations to make the results clearer at a glance.

Response: Thank you for your suggestion. We have updated the annotations in the figures based on your feedback to clearly display the statistical significance levels. We hope these changes make the results clearer at a glance. Please let us know if you have any further suggestions or need additional information.

15. Improve clarity in table legends by providing more context on how values (e.g., relative abundance, fold change) were derived

Response: Thank you for your feedback. We have updated the information in the tables to provide more background on the sources of the values. Now, readers can more clearly understand how values such as relative abundance and fold change are calculated and derived.

Reviewer #3 (Comments for the Author):

Dear authors you are doing well and that looks great work. A few issues, however, need to be addressed;

Response: We appreciate the anonymous reviewer for the nice comments on our efforts.

1. In line 25 Could the authors explain the specifics of the shotgun metagenomic sequencing method employed? Were any particular factors taken into account to guarantee precise fungal identification?

Response: For specific details on the metagenomic sequencing methods employed, please refer to section 2.3 "Processing of Metagenomic Sequencing Data" in the Materials and Methods. In our study, to ensure accurate fungal identification, we employed a specific approach: we first aligned the fungal genomes with the bacterial sequences in the NT database and removed any sequences matching the bacteria. We then generated a refined fungal-specific sequence set.

2. In line 27 How did you handle the difficulties associated with identifying temporary and commensal fungal species in the gut microbiome?

Response: Distinguishing between temporary and commensal fungal species in the gut microbiome poses a challenge with current technology. To enhance the scientific accuracy of our data, we implemented several strategies. Firstly, we removed potential dietary fungi to minimize interference in our analysis. Additionally, we employed advanced bioinformatics tools and databases to ensure the reliability of the identification process. Despite these efforts, the field is continuously evolving, and future technological advancements may further improve our ability to make these distinctions.

3. In line 29 What statistical methods were used to compare the fungal communities between ACVD patients and healthy controls?

Response: We are grateful for your focus on this question. We have used the Wilcoxon rank-sum test to statistically compare the fungal communities between ACVD patients and healthy controls. These results have been incorporated into the revised manuscript for clarity and comprehensiveness.

4. In line 60 The introduction, which mostly focuses on bacterial communities, highlights the role of the gut microbiota on ACVD. Could you explain why, in this specific scenario, the gut mycobiome has received very little research?

Response: When researching atherosclerotic cardiovascular disease (ACVD), the gut mycobiome has garnered less attention compared to bacterial communities, primarily due to several factors. Firstly, the gut mycobiome is more complex and difficult to understand, as fungi exhibit diverse roles and interactions that complicate comprehensive research. Secondly, analyzing fungal DNA and distinguishing between different fungal species is technically more demanding. The tools, techniques, and database information used to study fungi are not as advanced or comprehensive as those for bacteria. Thirdly, the direct impact of fungi on ACVD may not be as apparent or significant as that of bacteria. For example, bacterial metabolites such as TMAO have a well-established link to cardiovascular diseases, whereas similar clear connections involving fungi have yet to be established, thus lowering the research priority on fungi. Consequently, although the gut mycobiome is an important component of the overall microbiome, it has not been fully studied due to these combined factors.

5. In line 73 Why did you undertake a thorough research that included bacteria, fungi, and other microorganisms, instead of concentrating exclusively on fungal communities, given the intricate interactions between these two groups of microbes in the gut?

Response: The gut microbiome is a complex ecosystem where various microorganisms, including bacteria, fungi, viruses, and protozoa, interact

closely. Studying fungi in isolation may not provide a comprehensive understanding of the microbiome's overall function and its impact on host health. Additionally, the interactions between bacteria and fungi offer more extensive and detailed insights, allowing researchers to observe the connections between the gut microbiome and ACVD from multiple perspectives. Therefore, the study encompasses various microbial communities, including both bacteria and fungi, to capture these intricate dynamics and gain a more holistic view.

6. In line 76 *Malassezia* and *Candida* can alter immunological responses, according to the investigators. Are these fungi thought to affect the aetiology of ACVD through any particular mechanisms?

Response: We are grateful for your focus on this question. The study suggests that formyl-methionine, a metabolite of *Candida albicans*, activates intestinal HIF-2 α signaling, leading to increased ceramide synthesis and subsequently accelerating atherosclerosis. Additionally, *Malassezia* species, such as *Malassezia restricta*, may activate immune responses through the CARD9 pathway, resulting in systemic inflammation and the production of pro-inflammatory cytokines and chemokines. These immune responses could facilitate the formation of atherosclerotic plaques, thereby exacerbating the progression of ACVD. While these mechanisms are plausible and supported by existing biological pathways, further experimental studies are necessary to confirm these hypotheses. The relevant content is in lines 353-375 of the article.

7. In line 92 How were the subjects selected for the study? Were there any inclusion or exclusion criteria for the patients and healthy controls? Was the study prospective or retrospective?

Response: All patients were ethnic Han Chinese with no known consanguinity, aged 40–80 years old. The exclusion criteria included ongoing infectious

diseases, cancer, renal, or hepatic failure, peripheral neuropathy, stroke, as well as use of antibiotics within 1 month of sample collection. All the healthy control individuals enrolled were free of clinically evident ACVD symptoms at the time of the medical examination. Demographic data and cardiovascular risk factors were collected by a questionnaire. Individuals with peripheral artery disease, known coronary artery disease or myocardial infarction, cardiomyopathy, renal failure, peripheral neuropathy, systemic disease, and stroke were excluded. retrospective. The relevant content has been added to lines 99-108 of the article.

8. In line 93 Any preprocessing steps applied to the metagenomic data before analysis, such as quality filtering, adapter removal, and trimming.

Response: Thank you for your feedback. We have added the specific details regarding the preprocessing steps applied to the metagenomic data, including quality filtering, adapter removal, and trimming, to lines 141-148 in the manuscript.

9. In line 132 Give a thorough explanation of the normalization procedure that was utilized to determine the relative abundances. Talk about the genome size that was taken into account and if any further normalizing methods were used.

Response: Thank you for your insightful question. The normalization procedure used to determine the relative abundances involved initially adjusting the read count of each genome by its genomic size. This adjustment was important to account for differences in genome sizes among species. Subsequently, we applied the TPM normalization method, which involved dividing the normalized read count by the sum of all normalized read counts in the sample, and then scaling it to a per million basis. This comprehensive approach ensured accurate and comparable relative abundance estimations across samples.

10. In line 134 Before determining relative abundances, describe the normalization process used for the readings (e.g., TPM, RPKM, or other normalizing approaches).

Response: Thank you for your question. Before determining relative abundances, the normalization process for the readings employed the Transcripts Per Million (TPM) method. TPM is a widely used normalization approach that takes into account the length of the genome and the total read count in the sample. This method involves dividing the read count by the genomic size, and then by the total sum of all size-normalized read counts, scaling the result to a per million basis. This ensures that the relative abundances are comparable across different samples and conditions.

11. In line 185 Detailed information on the characteristics of the ACVD (atherosclerotic cardiovascular disease) patients and healthy controls (e.g., age, gender, lifestyle factors, medication use).

Response: Information on gender, age, and BMI can be found in Supplementary Table 1. Additionally, none of the ACVD patients had received steroids or antibiotics within the preceding three months.

12. In line 187 The authors report mild but significant separation between groups in PCoA. Is this separation biologically meaningful given the variance explained (18.4% and 10.4%)?

Response: Thanks for the reviewer for this question. In the gut microbiota, certain specific microorganisms can play a crucial role in maintaining gut health, immune regulation, and metabolic processes. Even if the overall change is small, it can lead to significant health issues. For example, bacteria such as *Faecalibacterium prausnitzii* that produce short-chain fatty acids (e.g., butyrate) may decrease in abundance, potentially affecting gut barrier function and anti-inflammatory actions, which are associated with inflammatory bowel disease (IBD). Therefore, the observed separation in the PCoA, although

explaining 18.4% and 10.4% of the variance, may still indicate a biologically meaningful shift in the microbial community. This could reflect underlying dysbiosis that contributes to disease pathology.

13. Does Table S3 provide enough information on the fold change and relative abundance of all significant genera? Is there a reason for choosing a cutoff of $FC > 1.2$ and relative abundance > 0.01 ?

Response: Table S3 contains detailed information on all the fungal genera that meet the selection criteria, including their significant fold changes and relative abundances. The selection criteria of $FC > 1.2$ and relative abundance > 0.01 ensure that we focus on changes with biological and statistical significance. An $FC > 1.2$ indicates at least a 20% change, filtering out minor changes that may be noise. A relative abundance > 0.01 ensures we study taxa with a significant presence in the microbiome, providing more reliable data. Together, these standards help identify taxa that have a more impactful and meaningful biological role.

14. In line 218 Are there known associations between the enriched genera (e.g., *Candida*, *Malassezia*, *Exophiala*) and ACVD pathophysiology? Could their enrichment be related to factors other than ACVD, such as diet, environment, or medication use?

Response: Studies suggest that fungi from the genera *Candida*, *Malassezia*, and *Exophiala* may be associated with the pathophysiology of atherosclerotic cardiovascular disease (ACVD). *Candida albicans* can accelerate the progression of atherosclerosis by activating the HIF-2 α signaling pathway in the gut, which increases ceramide synthesis. *Malassezia restricta* may trigger systemic inflammation through the CARD9 pathway, promoting the formation of atherosclerotic plaques. *Exophiala* fungi, as significant opportunistic pathogens, might impact cardiovascular health, although current research is limited and further investigation is necessary. Besides, we acknowledge that

diet and environment have certain impacts on the gut microbiome, but these impacts are filled with randomness. In our study design, we have made efforts to control these variables as much as possible and used appropriate statistical analysis methods to minimize their interference with the results. Nonetheless, to further confirm the specific mechanisms of these impacts, future research will delve deeper into this field to gain a more comprehensive understanding.

15. In line 257 Why was a random forest classifier chosen for this analysis? Were other machine learning models (e.g., support vector machines, logistic regression) considered or tested

Response: We have added the prediction results of the LASSO model. The specific content can be found in lines 321-325 and lines 398-402.

16. How was the random forest model tuned (e.g., number of trees, depth of trees) to optimize performance?

Response: Thank you for your valuable comments on our paper. Regarding the number of trees, we evaluated this by plotting the Out-of-Bag (OOB) error rate. Specifically, we gradually increased the number of trees and observed the changes in the OOB error rate. When we found that the OOB error rate gradually decreased and stabilized as the number of trees increased (see supporting figure below), we decided to choose 999 trees. This number was chosen because, at this point, the error rate showed no significant fluctuations, indicating that the model's performance had stabilized, and further increasing the number of trees would not significantly improve the model's performance.

17. In line 287 In what way does the discovery of elevated fungal diversity in patients with ACVD correspond with the current understanding of microbial dysbiosis in cardiovascular disorders? Exist conflicting results in the literature?

Response: Thanks for the reviewer for this question. The microbiota plays a crucial role in maintaining human health. Dysbiosis refers to the disruption of the microbial balance within the body and is commonly associated with various diseases, including cardiovascular diseases. This dysbiosis can manifest as a decrease in beneficial microbes or an increase in pathogenic ones, which in turn affects the host's metabolism and immune response, thereby promoting the development of CVD.

In current research on ACVD, it is observed that the overall diversity of gut fungi does not significantly change [1]. However, an increase in certain harmful fungi may be associated with the occurrence of ACVD, which aligns with the findings of this study. Therefore, the change in diversity does not necessarily contradict previous results, and this apparent discrepancy could be due to differences in sample size and methodology.

Reference: [1] K. An, Y. Jia, B. Xie, J. Gao, Y. Chen, W. Yuan, J. Zhong, P. Su, X.J.E. Liu, Alterations in the gut mycobiome with coronary artery disease severity, 103 (2024).

18. In line 295 Given the potential role of certain fungi in ACVD pathogenesis, are there any known antifungal treatments or dietary interventions that could be explored for therapeutic benefit?

Response: Thank you for your valuable suggestion. Currently, there are several antifungal medications, such as azole drugs like fluconazole and echinocandins, specifically used to treat infections caused by *Candida albicans* and *Malassezia* species. Additionally, probiotics like *Bifidobacterium* and *Lactobacillus*, and prebiotics such as fructooligosaccharides and galactooligosaccharides, may help restore microbial balance by promoting beneficial bacteria growth and inhibiting harmful fungi. However, their application in ACVD requires further clinical trials to confirm efficacy and safety, as the specific role of fungi in ACVD remains unclear. Therefore, these treatments and interventions have potential therapeutic benefits but need more research for validation.

19. In line 351 Outline specific future research avenues that could build on this study, such as investigating the functional roles of identified fungal taxa in ACVD, exploring the relationship between the mycobiome and metabolic risk factors (like TMAO), or conducting longitudinal studies to assess how gut mycobiome changes correlate with disease progression.

Response: In our future research, we plan to incorporate more metabolic data and clinical information, such as plasma TMAO levels and diabetes status. Using functional genomics and metabolomics, we will delve into the roles of identified key fungal taxa in gut function and metabolites, as well as the genomic sources of toxic substances. By analyzing the relationship between these metabolic risk factors and gut mycobiota, we aim to comprehensively understand the connection between the gut environment and ACVD risk, including key functional mechanisms, risk assessment of TMAO for ACVD, and intervention strategies involving fungal communities.

20. In line 298 To enhance the discussion section add the following reference: Al-Ouqaili, M.T.S., Musleh, M.H., Al-Kubaisi, S.M.A. Depending on HPLC and PCR, detection of aflatoxin B1 extracted from *Aspergillus flavus* strains and its cytotoxic effect on AFB treated-hematopoietic stem cells obtained from human umbilical cord. Asian Journal of Pharmaceutics, Volume 12, Issue 3, July-September 2018, Pages S1048-S1054.

In line 63 To enhance the introduction section add the following updated reference: Al-Moghira Khairi Al-Qaysi; Safaa Abed Latef Al-Meani; Mushtak T.S. Al-Ouqaili. (2020). The Effect of Dual-Species Biofilms, Monosaccharide and D-Amino Acids on Pseudomonas Biofilm. Indian Journal of Forensic Medicine & Toxicology, 15(1), 2177-2192. <https://doi.org/10.37506/ijfmt.v15i1.13728>

In line 70 To enhance the introduction section add the following update reference: Jomehzadeha, N, Javaherizadehd, H, Amin M, Saki M, Al-Ouqaili MTS, Hamidic H, Seyedmahmoudic M, Gorjiana Z. (2021). Isolation and identification of potentially probiotic Lactobacillus species from faeces of infants in southwest Iran. International Journal of Infectious Diseases 96: 524-530.

Response: Thank you for your suggestion. The citation reference has been added to the main text.

Re: Spectrum02182-24R1 (**Gut mycobiome alterations and network interactions with the bacteriome in patients with atherosclerotic cardiovascular disease**)

Dear Dr. Wei Yang:

Thank you for the privilege of reviewing your work. Below you will find my comments.

References format issue: journal names should be spelled correctly. For example:

ref38 should be: Liu B, Totten M, Nematollahi S, Datta K, Memon W, Marimuthu S, Wolf LA, Carroll KC, Zhang SX. J Mol Diagn. 2020. Development and Evaluation of a Fully Automated Molecular Assay Targeting the Mitochondrial Small Subunit rRNA Gene for the Detection of *Pneumocystis jirovecii* in Bronchoalveolar Lavage Fluid Specimens. 22:1482-1493.

ref39 should be: Liu BM. J Med Virol. 2024. Epidemiological and clinical overview of the 2024 Oropouche virus disease outbreaks, an emerging/re-emerging neurotropic arboviral disease and global public health threat.96:e29897.

Please return the manuscript within 7 days; if you cannot complete the modification within this time period, please contact me. If you do not wish to modify the manuscript and prefer to submit it to another journal, notify me immediately so that the manuscript may be formally withdrawn from consideration by Spectrum.

Revision Guidelines

Sincerely,
Benjamin Liu
Editor
Microbiology Spectrum

Responses to the Reviewer's comments

Dear editors and reviewers,

Thank you very much for your careful guidance. I have made the modifications in "Spectrum02182-24R1-Tables_S3" regarding the issues you raised.

Once again, thank you very much for your comments and suggestions.

Sincerely,

Wei Yang

Re: Spectrum02182-24R2 (**Gut mycobiome alterations and network interactions with the bacteriome in patients with atherosclerotic cardiovascular disease**)

Dear Dr. Wei Yang:

Your manuscript has been accepted, and I am forwarding it to the ASM production staff for publication. Your paper will first be checked to make sure all elements meet the technical requirements. ASM staff will contact you if anything needs to be revised before copyediting and production can begin. Otherwise, you will be notified when your proofs are ready to be viewed.

Sincerely,
Benjamin Liu
Editor
Microbiology Spectrum